# Identification and Analysis of Metabolites That Contribute to the Formation of Distinctive Flavour Components of Laoxianghuang

**DOI:** 10.3390/foods12020425

**Published:** 2023-01-16

**Authors:** Xi Chen, Liangjing Lin, Huitian Cai, Xiangyang Gao

**Affiliations:** 1Guangdong Provincial Key Laboratory of Nutraceuticals and Functional Foods, South China Agricultural University, Guangzhou 510630, China; 2SCAU (Chaozhou) Food Institute Co., Ltd., Chaozhou 521000, China

**Keywords:** bergamot, Laoxianghuang, untargeted metabolomics, differential metabolites, flavour

## Abstract

In addition to volatile compounds, metabolites also have a great effect on the flavour of food. Fresh finger citron cannot be eaten directly because of its spicy and bitter taste, so it is made into a preserved fruit product known as Laoxianghuang (LXH). To investigate the metabolites that have an effect on the flavour of LXH, untargeted metabolomics was performed using an ultrahigh-performance liquid chromatography with tandem mass spectrometry (UPLC-MS/MS), and the metabolites of the Laoxianghuang samples from different locations in the Chaoshan area were compared and analysed. A total of 756 metabolites were identified and distinct differences were revealed among the different Laoxianghuang samples. A total of 33 differential metabolites with the most significant changes were screened through further multivariate analytical steps, and each group of samples had unique metabolites. For instance, pomolic acid had the highest content in the JG sample, while L-glycyl-L-isoleucine was rich in the QS sample. Moreover, flavonoid metabolites made the greatest contribution to the unique flavour of Laoxianghuang. The metabolic pathways involved are the biosynthetic pathways of flavonoids, isoflavonoids, flavones, and flavonols. This study can provide some creative information for distinguishing the quality differences of Laoxianghuang from the perspective of metabolites and offer preliminary theoretical support to characterise the formation of flavour substances in Laoxianghuang.

## 1. Introduction

In recent years, people’s dietary demand has increased as their standards of living have developed. Consumers pay attention not only to their foods’ nutritional value but also to their palatability and their unique flavour, with the result that food aroma has become increasingly popular by virtue of its multifunctional nutrient value and its distinctive flavour generated from microbial metabolism [1,2]. During fermentation, microorganisms decompose available carbohydrates into certain substances, such as organic acids, water, and carbon dioxide, as well as other metabolites, which have a great impact on the aroma, flavour, and taste of food [3]. A significant correlation between the microbiome (bacterium and fungi) and volatiles in Xiangxi sausages was confirmed [4]. Apart from fermented food, unfermented food similarly varies in flavour and appearance because it is rich in bioactive metabolites. In previous studies, the flavour of fruits and vegetables is greatly influenced by the concentrations of carbohydrates, organic acids, and polyphenols, and the taste can be affected by amino acids [5]. The sweet flavour in meat originates from the presence of sugars, amino acids, and organic acids, and the sour flavour emerges after amino acids are coupled with organic acids. Inorganic and sodium salts of glutamate and aspartate generate salty taste, while bitter taste is probably due to hypoxanthine, anserine, and carnosine, as well as some amino acids [6]. These findings imply that the highly bitter taste of Kucha may be due to the combined effects of catechins, alkaloids, flavonols and flavonol/flavone glycosides, amino acids, and phenolic acids [7]. It can be concluded that, in addition to volatile compounds, nonvolatile compounds and other chemicals derived from their own or microbial metabolism can lead to significant changes in food flavour [8].

Bergamot (*Citrus medica* L. var. sarcodactylis Swingle), widely known as “Foshou” in China, is the fruit of evergreen shrubs or small trees of the Rutaceae. It is used as a traditional medical food with the functions of expectorating phlegm; relieving cough and asthma; having anti-inflammatory, anti-bacterial, anticancer, and anti-depressive properties; and lowering blood pressure [9]. Bergamot contains a variety of nutrients, such as polysaccharides, flavonoids, terpenoids, and coumarin compounds, and has a variety of biological activities, such as bacteriostatic properties [10]. Because of its unique aroma, bergamot is often refined into essential oil, which is added to food in small amounts as flavourants and is in great demand in the cosmetic and perfume industries [11]. GC-MS has shown that the volatile component of fresh bergamot is mainly limonene and is more pronounced in the mature stage with α-thujone, 3-carene, α-pinene, β-pinene, γ-terpinene, monoterpene hydrocarbons, and ketones [12]. While its taste is too spicy and bitter, and hence it cannot be eaten directly, only a small amount needs to be included as an additive and is usually applied to fermentations. Laoxianghuang (LXH) is made from fermented bergamot, which requires a series of complicated pretreatment processes, including pickling, sugaring, desalination, cooking and drying, and fermentation. The fermentation time of LXH to produce an excellent flavour needs to be at least three years [13]. Fermentation not only improves the original unique flavour of bergamot but also retains its nutrients and related medicinal effects. The obtained Laoxianghuang has a richer and more mellow flavour (sweet fragrance with slight saltiness), with the effects of soothing the liver, regulating gas, relieving pain in the stomach, eliminating dampness, and resolving phlegm. It is traditionally considered that the longer LXH is aged, the sweeter the fragrance and the mellower the flavour are. Electronic nose, GC-MS, and GC-IMS have been utilized to analyse the changes in the volatile components in LXH during fermentation, indicating that the volatile components of LXH, dominated by trans-orange flower tertiary alcohol, citronellol, 2-ethylfuran, etc., begin to change markedly after six months of fermentation. Moreover, LXH manifests a wood odour from the 3rd to the 10th year of fermentation, while a herb odour is exhibited in the 15th and 20th years, indicating that the fermentation time has great influence on the composition of volatile flavour substances in Laoxianghuang [14]. Similar to LC-MS and Q-TOF-MS, ultrahigh-performance liquid chromatography–tandem mass spectrometry (UPLC-MS/MS) has the features of fast analysis, high sensitivity, and strong anti-interference ability. In complex sample analysis, it can effectively reduce matrix interference and improve the accuracy of detection. It is suitable for the simultaneous determination of various complex major and trace components in samples [15]. Metabolomics is the science of information integration and biomarker identification through high-throughput detection and data processing of low-molecular-weight metabolites (such as organic acids, fatty acids, amino acids, and sugars) in biological samples, which can detect dozens or even hundreds of endogenous metabolites and has been widely used in food research [16]. Untargeted metabolomics can reflect the changes in metabolites as a whole, which is beneficial to the discovery of new metabolic pathways. Previous studies have demonstrated significant advances in metabolomics in sample authentication based on geographical origin [17,18] and variety [19,20], monitoring dynamic changes in experimental and production processes [21,22], and analysing metabolite correlations with food sensory characteristics [23,24]. The production of LXH is currently still dominated by small family businesses, with few standardised and professional enterprises, which leads to uneven product quality. The lack of a unified quality judgement standard seriously hinders the promotion of LXH in the market. There has been no relevant report on the use of metabolomics to analyse the effect of metabolites on the characteristic flavour components of LXH in existing research. Therefore, in the present investigation, untargeted metabolomics based on UPLC-MS/MS was conducted to screen and analyse differential metabolites that can result in the distinctive flavour of LXH and their relevant biosynthetic pathways, and to explore the correlation between flavour and metabolites to provide a reliable theoretical guidance for the development of quality evaluation criteria of LXH.

## 2. Materials and Methods

### 2.1. Collection of LXH Samples

The samples for the experiment were all LXH with 1 fermentation year, which were randomly collected from different brands in the Chaoshan area of Guangdong Province (China) and were representative. A total of 12 samples were randomly collected in this experiment and were divided into 4 groups according to the brands, named JG, QS, YS, and CK. Each group separately contained 3 biological replicates. JG1, QS1, YS1, and CK1 represented every type of LXH collected that was made of Guang berbamot (one of the bergamot varieties) and had been fermented for 1 year. The samples were transported to our laboratory and stored in a cool and dry place at room temperature.

### 2.2. Metabolite Extraction from Samples

Methanol and acetonitrile were purchased from Merck (Guangzhou, China). The dimethyl sulfoxide standards used for calibration were obtained from Sigma-Aldrich and Bio BioPha (Guangzhou, China). All chemicals and reagents used were chromatographically pure.

Briefly, the LXH samples were placed in a lyophiliser to vacuum freeze-dry them (Scientz-100F). Three replicates were prepared for each sample. A grinding mill was utilised to grind the sample to powder, and 100 g powder was dissolved in 1.2 mL of 70% methanol. The methanol extraction was vortexed every 30 min for 30 s each time, which was repeated 6 times. The extraction solution was placed in a 4 °C refrigerator overnight and then centrifuged at 12,000 rpm for 10 min. The supernatants were filtered through a 0.22 μm microfiltration membrane, transferred to glass vials, and analysed using UPLC-MS/MS.

### 2.3. UPLC-MS/MS Analytical Conditions

Metabolite measurements were performed using a UPLC-MSMS/MS system. SHIMADZU Nexera X2 (Shimadzu, Tokyo, Japan) was used for ultrahigh-performance liquid chromatography, which was used in tandem with the Applied Biosystems 4500 QTRAP mass spectrometer(Applied Biosystems, New York, NY, USA). UPLC was performed on an SC-18 column (2.1 mm × 100 nm, Agilent Technology, Santa Clara, CA, USA) with a solvent flow rate of 0.35 mL/min at a column temperature of 40 °C. The injection volume was 4 μL. The mobile phases included A (ultrapure water containing 0.1% formic acid (*v*/*v*)) and B (acetonitrile containing 0.1% formic acid (*v*/*v*)). The elution gradient of the mobile phase was as follows: 0 min 5% B; B increased linearly up to 95% within 9 min; B was maintained at 95% for 1 min; the ratio of B dropped to 5% until 10–11.1 min; and B remained at 5% until 14 min.

Linear ion hydrazine (LIT) and triple quadrupole rod (QQQ) scans were obtained from a mass spectrometer (Applied Biosystem 4500 QTRAP, USA) applied for MS analysis, which was equipped with an ion spray interface (ESI Turbo). The operation of positive and negative modes could be controlled by software (Analyst 1.6.3, ABSciex, Singapore). The following were the parameters of the ESI operation: ion source; turbine spray; source temperature at 550 °C; ion spray voltage at 5500 V(+)/−4500 V(−); and ion source gas I (GSI), ion source gas II (GSII), and curtain gas (CUR) set to 50, 60, and 25 psi, respectively. The collision-induced power parameter was set to high. Polypropylene glycol solutions (10 μmol/L and 100 μmol/L) were used for instrument tuning and quality calibration in the QQQ and LIT modes, respectively. The QQQ scanning mode was MRM, and the collision gas (nitrogen oxide) was set to the medium level. Through further optimisation of declustering potential (DP) and collision energy (CE), the DP and CE of each MRM ion pair were completed. A specific set of MRM ion pairs was monitored at each period based on the metabolites eluted during each period.

### 2.4. Data Processing and Metabolite Recognition

According to the secondary spectral information, the material was qualitatively analysed, and the isotope signals, some ions (including K^+^, Na^+^, and NH4^+^), and other ions with a larger molecular weights were removed based on a self-built database—metware database (MWDB). The metabolite quantification was a multiple reaction monitoring (MRM) model using triple quadrupole mass spectrometry. In the MRM mode, the quadrupole first screened the precursor ions of the target substance, and the ions corresponding to other molecular weight substances were excluded to preliminarily eliminate interference. The precursor ions were induced by the collision chamber to ionise and break to form many fragment ions. The fragment ions were filtered by the quadrupole to choose a required characteristic fragment ion so that interference could be further eliminated. After obtaining the metabolite spectral data of different samples, the mass spectral peak areas of all substances were integrated, and the mass spectral peak of one metabolite in different samples was corrected by integration.

### 2.5. Multivariate Data Analysis

The metabolite data were processed, and the total ion current (TIC) coupled with the MRM multipeak map of metabolite detection (XIC) were attained using the Analyst 16.3 software. The data were imported into the SIMCA14.1 software (Umetrics, Umeå, Sweden) for multivariate statistical analysis, including principal component analysis (PCA), partial least squares discrimination analysis (PLS-DA), orthogonal partial least squares discrimination analysis (OPLS-DA), and variable importance in projection (VIP). The results of the PCA showed the separation trends of metabolites among the groups, indicating whether there were differences in metabolites among the sample groups. Based on the results of the OPLS-DA, the metabolites with differences between different varieties or tissues could be preliminarily screened from the VIP, and the differential metabolites could be further screened by combining the *p*-value or the fold change. Metabolites showing significant differences were considered to have a fold change ≤0.5 (≥2.0) and VIP ≥ 1.0. In addition, the intersection of all significantly different metabolites and the expression levels of different metabolites were taken to perform hierarchical clustering analysis (HCA) to intuitively indicate the relationship among the samples and the expression differences of metabolites among different samples. The metabolite content data were normalized by unit variance scaling (UV), and a heatmap was created using the Origin 2018 Pro software.

### 2.6. KEGG Pathway Enrichment Analysis of Differential Metabolites

The KEGG pathway database was used to annotate the detected differential metabolites and to perform pathway enrichment analysis. The differential metabolites were reflected in the KEGG data to identify their KEGG ID and the pathway to which they belong. The number of metabolites enriched in the corresponding pathway was then counted. The *p*-value was used to determine whether the pathway was enriched or not, and it was considered to be enriched when the *p*-value was ≤0.05.

## 3. Results

### 3.1. Identification of Metabolites

A total of 764 nonvolatile metabolites, including 341 negative-ion-mode metabolites (ESI−) and 423 positive-ion-mode metabolites (ESI+) were identified. These metabolites mainly consisted of 12 categories of metabolites, including 54 organic acids, 67 phenolic acids, 94 lipids, 280 flavonoids, 62 amino acids and their derivatives, 40 lignans and coumarins, 20 nucleotides and their derivatives, 32 alkaloids, 32 terpenoids, 8 tannins, 1 steroid, and 74 other categories of metabolites. The relative percentages of the metabolite components are detailed in Figure 1. The content of each metabolite is expressed by relative abundance.

Based on the identification results of the metabolites, lipids (33–42%), flavonoids (8–24%), organic acids (3–19%), phenolic acids (9–13%), and amino acids and their derivatives (3–9%) were the main non-volatile metabolites in LXH, which showed that the contents of flavonoids and organic acids varied greatly among the groups of LXH, while the content of lipids and phenolic acids had a smaller range of difference. In addition, most volatile components, such as terpenoids, arenes, aldehydes, and alcohols, were not detected. These results were consistent with previous research showing that the volatile components of finger citron samples at different pickling stages included terpenoids, arenes, alcohols, phenols, aldehydes, esters, acids, ethers, ketones, and others, among which terpenoids, arenes, phenols, and aldehydes were the major components, which greatly contributed to the fragrance of finger citron [25]. Furthermore, the contents of flavonoid and polyphenol were significantly lower than those in the fresh bergamot, suggesting that large proportions of such substances had been lost in the salting stage [13]. The method of ordinary one-way ANOVA was used to analyse the data (Figure 1B). The total metabolite content was significantly different between all groups of different LXH samples (*p* < 0.05).

### 3.2. Multivariate Statistical Analysis for Differential Metabolites

The metabolite components detected in the LXH samples from different regions are presented in the PCA score scatter plot shown in Figure 2. The first two components explain 43.8% (PC1) and 27.2% (PC2) of the total variance, respectively (R2X = 0.893), indicating that the two principal components contribute to the primary characteristic information of the different samples. The compact distance of the points also indicates good repeatability in the same groups. In addition, the QS and CK samples are closely spaced in a small area, suggesting that the two groups of samples have similar main components and that their differences in metabolites are not obvious. The results also reveal that JG, YS, and CK (or QS) have significant variation since they are dispersed in three different regions in the PCA model. Aurapten, 1-O-feruloyl-D-glucose, 2′-hydroxygenistein, pinocembrin-7-O-rutinoside, sudachiin C, and licochalcone C have the most positive contributions to PC1, while gluconic acid, N-glycyl-L-leucine*, 2-furoic acid, 2-methylglutaric acid, and L-glycyl-L-isoleucine* are most positively correlated with PC2 (Figure 2).

The OPLS-DA model was further established to demonstrate the differences in non-volatile metabolites in the different LXH samples. In general, R^2^Y provides an estimate of how well the model fits the Y data, whereas Q^2^ is an estimate of how well the model predicts the Y data. To achieve high predictive ability, the values of R^2^Y and Q^2^Y should be close to one. As illustrated in Figure 3A, it is obvious that all samples are spatially separated and well distinguished, particularly between JG and YS; in contrast, the level of metabolite difference between QS and CK is smaller. As shown in the OPLS-DA score plot (Figure 3A), a clear discrimination of the samples from different sources is achieved, suggesting a strong relationship with the profile of secondary metabolites. The OPLS-DA model generates values of 0.993 and 0.985 for R^2^Y and Q^2^Y, respectively, suggesting that the model has excellent reliability and predictability. No outlier samples could be observed by Hotelling’s T2. The results of the OPLSDA model were verified by a permutation test (R^2^ = (0.0, 0.14), Q^2^ = (0.0, 0.84)); the Q^2^ point of the model from left to right is much lower than the original Q^2^ point, and the R^2^ and Q^2^ values of the model are more than 0.9. The intercept of the Q^2^ regression line is −0.479, which indicates that the model can reliably predict the results without any overfitting phenomenon (N = 100).

The corresponding S-plots based on the OPLS-DA model were constructed to visualise the relationship between the covariance (p[1]) and the correlation (p (corr)) of the principal components and metabolites, in which the metabolite markers closest to the bottom left and top right are considered to have the most significant variance. The metabolites with VIP values greater than or equal to one are marked by red dots, whereas those with VIP values smaller than one are marked by green dots in Figure 3B–H. From the S-plots of pairwise sample comparisons, the red dots closest to the bottom left and top right are citric acid, succinic acid, L-malic acid, muconic acid, melianone, 5,7-dimethoxycoumarin (Limettin) (citropten), and stachydrine, of which citric acid is the most common metabolite, with the most significant difference among the pairwise samples.

### 3.3. Classification and Screening of Differential Metabolites

The variable importance in projection (VIP) values, following the supervised OPLS-DA model, reflect the degree of influence of the difference between groups of corresponding metabolites in the classification and discrimination of samples in the model; the VIP values were applied to further identify and screen for differential metabolites. Metabolites with VIP values ≥1 are usually regarded as significantly different metabolites.

To explicitly clarify the influence of different regions on the metabolites of all samples and to further screen for differential metabolites, volcano plots of comparisons between pairs of samples were performed, as shown in Figure 4. More concretely, according to the principle of VIP values > 1 and fold change ≥2 or ≤0.5 for screening differential metabolites, a total of 2239 different substances are screened (Figure 4A–F), out of which 314 differences are recorded between YS and CK (Figure 4A, showing 47 metabolites of upregulation and 47 metabolites of downregulation); 391 differences are observed between YS and QS (Figure 4B, showing 363 metabolites of upregulation and 28 metabolites of downregulation); 403 differences are observed between YS and JG (Figure 4C, showing 331 metabolites of upregulation and 72 metabolites of downregulation); 321 differences are marked between CK and QS (Figure 4D, showing 203 metabolites of upregulation and 118 metabolites of downregulation); 392 differences are marked between CK and JG (Figure 4E, showing 179 metabolites of upregulation and 213 metabolites of downregulation); and 371 differences are marked between JG and QS (Figure 4F, showing 217 metabolites of upregulation and 154 metabolites of downregulation). The differential metabolites are classified into several categories. In these categories, the majority of differential metabolites are flavonoid compounds, lipid and lipid-like molecules, and amino acids and their derivatives. More than half of the flavonoid metabolites, including flavonols, flavonones, isoflavones, flavones, and chalcones, are upregulated, except in the YS vs. CK and JG vs. CK comparisons. It is worth noting that, compared to CK, the flavonoid metabolites of YS and JG groups are downregulated, whereas they are upregulated in QS. There are 172 and 126 downregulated flavonoid metabolites in the YS vs. CK and JG vs. CK comparison groups. More than half of the flavonoid compounds show upregulation in comparison to YS. Furthermore, almost all lipid metabolites are downregulated, except for 7S,8S-DiHODE (9Z,12Z)-(7S,8S)-dihydroxyoctadeca-9,12-dienoic acid and methyl 7,10-hexadecadienoate between YS and CK. Only lysoPC 18:0 (2n isomer) shows downregulation between QS and YS. In contrast to JG, there are more varieties of amino acids and their derivatives in YS and CK, which show upregulation other than N-acetyl-L-glutamic acid. In the YS vs. CK group, only nine amino acids and derivatives are considered significantly different, of which only L-methionine and 2,3-dimethylsuccinic acid are downregulated.

As shown in the Venn diagrams of Figure 5A,B, not only common metabolites but also some characteristic metabolites exist between the different comparison groups after taking the intersections of each comparison group. A total of 157 common metabolites are observed among the CK vs. YS, CK vs. QS, and JG vs. CK comparison groups. In addition, the same 150 metabolites are shared among the QS vs. JG, QS vs. YS, and JG vs. YS comparison groups. Some differential metabolites are also found. These results indicate that the differential metabolites that cause differences can be vastly different and be applied to distinguish different Laoxianghuang samples.

Based on the VIP values >1 and fold change ≥2 or ≤0.5 mentioned above, Log2 (fold change) values of differential metabolites were sorted, and the top 10 metabolites with the highest different multiples (upregulated and downregulated) in each pairwise sample are listed in Table 1. The top 10 metabolites with the highest different multiples are 33 in total, including 14 flavonoids, 4 lipids, 4 other compounds, 3 amino acids and their derivatives, 4 acids (phenolic acids and organic acids), 4 alkaloids, 1 tannin, and 1 triterpene. The metabolites with the largest upregulation differences in each comparison group are rhoifolin, rhoifolin, 6-O-caffeoylarbutin, adipic acid, rhoifolin, and rhoifolin, respectively. Meanwhile, licorisoflavan A, adipic acid, N-glycyl-L-leucine, brevifolin carboxylic acid LysoPE 18:3(2n isomer), and licoagrochalcone D are the metabolites with the largest downregulated differences. Flavonoids are mainly upregulated by large degrees, while lipids show large downregulations. Notably, the same substances show the opposite states of expression in different pairwise samples. For instance, licoagrochalcone D is upregulated in CK vs. YS, while it is downregulated in CK vs. JG. Ellagic acid-4-O-xyloside is upregulated in CK vs. QS, while it is downregulated in CK vs. YS.

### 3.4. Hierarchical Cluster Analysis

A heatmap with hierarchical cluster analysis was applied to characterise the distribution of different metabolites among the LXH samples as shown in Figure 6. The abscissa represents different experimental groups; the ordinate represents different metabolites between the two groups; the colour blocks at different positions represent relative expression levels of metabolites at the corresponding positions; red represents upregulation of metabolite content; and green represents downregulation of metabolite content. The heatmap indicates that some metabolites, such as orientin glycosides and luteolin glycoside, show obviously higher contents in QS and CK than in YS and JG, whereas the contents of licoflavonol, pomolic acid, gancaonin D, licorisoflavan A, and 7-methyllicoricidin show the highest expression only in JG. The only triterpene derived from ursolic acid, pomolic acid, was first isolated from the peels of apples in 1966 by Brieskorn. Although some plants, such as coco plum (*Chrysobalanus icaco*), loquats (*Eriobotrya japonica*), and rosemary, are widespread, their pomolic acid contents are usually very low [26]. Mixtures of tormentic and euscaphic acid can be used to obtain pomolic acid using a convergent synthetic approach, which is considered to be the best and shortest route to pomolic acid generation [27]. Previous studies in vivo have demonstrated that pomolic acid isolated from Licania pittieri has a hypotensive effect [28] and is also capable of initiating apoptosis of ovarian carcinoma and inhibiting leukocyte growth [29]. 6-Hydroxyhexanoic acid, 4-[1-hydroxy-2-(methylamino)ethyl]phenol, Lyso PC17:2, and phenethylamine show the highest expression in QS. Moreover, lipid and lipid-like substances mainly accumulate in QS and the least in YS, while flavonoid metabolites mainly accumulate in YS and the least in QS. According to the information above, evident species and contents of metabolite differences exist among the LXH samples, which have possibly resulted from each kind of manufacturer modifying the raw material and crafting their product on the basis of the traditional formula.

### 3.5. Correlation Analysis of Metabolites

In this study, biological duplication between the samples within a group can be observed; the higher the correlation coefficient of the samples within a group compared to the samples between groups, the more reliable the differential metabolites obtained. As shown in Figure 7A, it is clear that there is excellent biological repeatability within the same group. There are all positive correlations between different LXH samples. Specifically, much stronger positive correlations exist between QS and YS, whereas CK shows the lowest positive correlations compared to any other groups. In addition, the correlation analysis was used to reveal the mutual relationships of these 33 differential metabolites, as shown in Figure 7B. The shapes and colours of the dots represent the correlation coefficients between the metabolites. In summary, 1089 correlation pairs were analysed, among which 275 metabolic pairs result in highly significant correlations (*p* < 0.01). Among the 275 correlation pairs, 206 positive correlations (r > 0, *p* < 0.01) and 69 negative correlations (r < 0, *p* < 0.01) are observed. The correlation pairs were further screened for extremely strong correlations and significance (|r| > 0.9, *p* < 0.001). For instance, licoflavonol, 6-O-caffeoylarbutin, licoagrochalcone D, licoagroside B, and 4′,5-dihydroxy-3′,5′-dimethoxy flavone are strongly positively correlated with licoarylcoumarin, while chrysin (r = −0.970) is strongly negatively correlated with luteolin-6-C-arabinoside-7-O-glucoside, and orientin-7-O-arabinoside and luteolin-7-O-glucoside-5-O-arabinoside are highly positively correlated with isolicoflavone B and apigenin-7-O-neohesperidoside. It was reported that apigenin could significantly inhibit UV-induced mouse skin tumorigenesis, and luteolin exhibited a high 2,2-diphenyl-1-picrylhydrazyl scavenging activity, with naringenin having a similar effect [30]. In addition, Mekawy found that apigenin, as an important intermediate flavonoid metabolite, could significantly enhance the adaptation of rice seedlings to salinity [31].

Some correlation pairs also show extremely positive correlations (r = 0.999), including isoglyasperin D with licoflavanol and 4′,5-dihydroxy-3′,5′-dimethoxyflavone and licoarylcoumarin with isoglyasperin D and glabrene, respectively. N-Glycyl-L-leucine is strongly positively correlated with only L-glycyl-L-isoleucine, whereas it shows highly negative correlations with rubrofusarin-6-O-glucoside, naringenin-4′-O-glucoside, isochlorogenic acid A, cutin-7-O-glucoside, 6-O-caffeoylarbutin, persicoside, and syringesinol-4′-O-glucoside.

### 3.6. KEGG Enrichment Analysis

Differential metabolites interact in organisms to form different metabolic pathways. Pathway enrichment analysis of differential metabolites is helpful to understand the mechanisms of metabolic pathway changes. The numbers of metabolic pathway diagrams with significant differences (*p* < 0.05) are nine, five, five, three, eight, and three, respectively as shown in Figure 8. The pathways of differential metabolites in groups JG and CK are primarily concentrated in piperidine and piperidine alkaloid biosynthesis, starch and sucrose biosynthesis, and purine biosynthesis. All groups, with the exception of JG vs. CK, are mainly concentrated in the biosynthesis of isoflavonoids, flavonoids, flavones, and flavonols. The differential metabolites involved in flavonoid, flavone, and flavonol biosynthesis are primarily isoliquinitigenin, butein, butin, 7,4-dihydroxyflavone, naringenin, hesperetin, garbanzol, phlorizin, neohesperetin, apigenin, luteolin, tricetin, etc. Flavonoid biosynthesis begins with the amino acid phenylalanine, from which phenylpropanoids are produced and enter the flavonoid–anthocyanin pathway. Naringenin is generated by chalcone isomerase after a multistep reaction, and naringenin indirectly participates in the isoflavone biosynthetic pathway. Apigenin generated by naringenin under the catalysis of flavone synthases I and II can indirectly participate in flavone and flavonol biosynthesis. One of the intermediates of this pathway, kampferol, can also indirectly participate in the flavonol biosynthetic pathway [32,33]. Naringenin is also the precursor for eriodictyol biosynthesis by flavanone 3′-hydroxylase catalysis, as well as pentahydroxyflavanone biosynthesis by flavanone 3′,5′-hydroxylase catalysis [34]. Flavone biosynthesis is a branch of the flavonoid biosynthetic pathway, in which flavone synthase catalyses the conversion of flavanones to flavones, such as apigenin, dihydroxyflavone, luteolin, and tricetin. Flavanones can also be converted to apigenin C-glycosides and luteolin C-glycosides by flavanone-2-hydroxylase [35]. As an intermediate product, chalcone is involved in a variety of biosynthetic pathways, including flavonone, flavone, isoflavone, and flavonol biosyntheses [36]. In the cytoplasm, chalcone isomerase participates in the cyclisation of chalcones to produce flavanones, opening a route to heterocyclic C-ring-containing flavonoids [37].

## 4. Discussion

### 4.1. Differential Metabolites Related to Flavonoids

Flavonoids are a class of major secondary metabolites that play crucial roles in plant tolerance to environmental stress. In addition, extracted flavonoids have been proven to be beneficial, with anti-diabetic, anti-inflammatory, and anti-fatigue properties [38]. Similar to other citrus plants in Rutaceae, flavonoids are the most abundant bioactive components in fresh finger citron. In general, most natural flavonoids in plants are bitter and astringent, but their tastes vary depending on their structures. Major citrus flavonoids, hesperidin and narirutin, have been reported to be tasteless. Although neohesperidin and naringin are structurally similar to hesperidin and narirutin, respectively, they have a strong bitter taste. Moreover, some dihydrochalcones (e.g., neohesperidin dihydrochalcone) and their derivatives have a strong sweet taste and are widely used as bitterness inhibitors in food [39]. Bitterness in fresh bergamot mainly results from hesperidin, naringin, and other citrus glycosides with flavanone-7-O-new orange peel glycoside structures. In the metabolites previously determined, the bitterness substances of the LXH samples mentioned above were downregulated and not the most significant, which indicated that these substances were inhibited after the fresh bergamot was preserved. Six flavonoid glucosides were screened, including 2 orientin glycosides (orientin-7-O-arabinoside and orientin-2″-O-xyloside), 1 kaempferol (kaempferol-3-O-(6″-p-coumaroyl)glucoside), 1 apigenin glycoside (apigenin-7-O-neohesperidoside), and 2 luteolin glycosides (luteolin-6-C-arabinoside-7-O-glucoside and luteolin-7-O-glucoside-5-O-arabinoside). A previous report revealed that flavonoid metabolites, such as hesperetin and limettin, play crucial roles in the formation of the special flavour of tangerine peel (Citri Reticulatae Pericarpium). With the extension of age, the aroma of tangerine peel becomes more mellow, and the content of flavonoids increases [40]. It is speculated that flavonoid metabolites have a positive effect on the special aroma of Laoxianghuang. Aside from the contribution to the special flavour, it has been reported that flavonols and flavonol glycosides, such as flavonol-3-glycosides, contribute to the astringency of black and green tea at low-threshold concentrations and to the bitterness of tea infusions. Twenty-six differential flavonols and flavonol glycosides were identified in the five analysed tea cultivars, with most being associated with astringency [41]. The flavonols and flavonol glycosides detected, such as licoflavonol and kaempferol-3-O-(6″-p-coumaroyl)glucoside, differentially contribute to the bitterness and astringency of Laoxianghuang, whereas their species and content are less than those of fresh bergamot.

### 4.2. Differential Metabolites Related to Lipids and Lipid-like Molecules

The lipids in food are not only responsible for energy, texture, and mouthfeel, but they also significantly contribute to the development of both desirable and undesirable odours and flavours by generating volatile compounds, acting as precursors for odour and flavour compounds (e.g., alcohols, aldehydes, and ketones) or modifying the odour and flavour of other compounds [42]. Among the 33 differential metabolites, the various lipid metabolites examined included one free fatty acid, glycerides, one lysophosphatidylcholine, and two lysophosphatidylethanolamines, and all showed downregulation trends. It is inferred that some phospholipids in the samples are degraded to free fatty acids and lysolecithin to a certain extent. Furthermore, phosphatidylcholine and phosphatidylethanolamine have been found to stimulate the formation of carbonyl metabolites, including hexanal, 2,4-dienal and octen-3one, under heating [8], thereby indicating that phospholipid degradation is significantly related to aroma formation in Laoxianghuang.

### 4.3. Differential Metabolites Related to Phenolic Acids and Organic Acids

Plant polyphenols, a class of nonvolatile secondary metabolites of plant origin, apart from their sensory properties of bitterness, also exhibit antioxidant, lipid-lowering, anti-inflammatory, anti-tumour, and cardiovascular prevention activities and are important ingredients for the preparation of functional foods [43,44]. In addition to removing lipid-derived carbonyl compounds that are essential for food flavour, phenolic compounds can also be converted enzymatically or nonenzymatically into quinones, which are able to degrade amino acids and can have significant impacts on food flavour [45]. Ceccaroni in 2018 found that polyphenols in beer can absorb 65% of the bottleneck oxygen, which contributes to the taste and flavour stability of beer [46]. Wannenmacher believed that the effect of polyphenols on the flavour stability of beer depends on the type and amount of flavonoid compounds [47]. Green tea’s bitter and astringent properties are primarily attributed to some typical phenolic acids, such as gallic acid (GA), chlorogenic acid (CGA), and caffeic acid (CA), which impart a sour and astringent taste that increases with the phenolic acid concentration [48]. It has been confirmed that organic acids contribute to the sourness and the fruity taste of food. In addition, it has been found that most sensory variation in fruit acidity has been linked to the sugar/acid content of fruit [49]. Of the 33 differential metabolites screened, 6-O-caffeoylarbutin, brevifolin carboxylic acid, and adipic acid had the greatest increases, while brevifolin carboxylic acid was significantly downregulated, affecting the flavour of LXH.

## 5. Conclusions

This study used untargeted metabolomics analysis based on UPLC-MS/MS to identify and screen for differential metabolites from four kinds of Laoxianghuang samples collected from different regions in the Chaoshan area. A total of 756 metabolites, including 280 flavonoids, 94 lipids, 67 phenolic acids, 62 amino acids and their derivatives, 54 organic acids, 40 lignans and coumarins, 32 alkaloids, 32 terpenoids, 20 nucleotides and their derivatives, 8 tannins, 1 steroid, and 74 other categories, were identified. Distinct differences were revealed among the different LXH samples, with 33 differential metabolites; the most significant changes were screened through further multivariate analytical steps, and each group of samples was found to have unique metabolites. For instance, pomolic acid had the highest content in JG, while L-glycyl-L-isoleucine was rich in QS. It is possible that different specific production formulations and manufacturing processes are responsible for the differences in metabolite composition among the samples. The unique flavour of Laoxianghuang is the result of the interaction of various metabolites, especially flavonoids, phenolic acids, and organic acids. Moreover, flavonoid metabolites make a great contribution to the unique flavour of Laoxianghuang. This study can provide some creative information for distinguishing the quality differences of Laoxianghuang from the perspective of metabolites, which could be utilised in quality control and offer novel insights into the characterization of the formation of distinctive flavour substances of Laoxianghuang. This study investigated the changes in non-volatile metabolites of different brands of LXH and related metabolic pathways, but there are still some limitations. Correlation analysis with microorganisms, volatile components, and other indicators can be conducted in subsequent studies. Further analysis of the functions of the individual microorganisms in these pathways will be beneficial to elucidate the metabolic mechanism.

## Figures and Tables

**Figure 1 foods-12-00425-f001:**
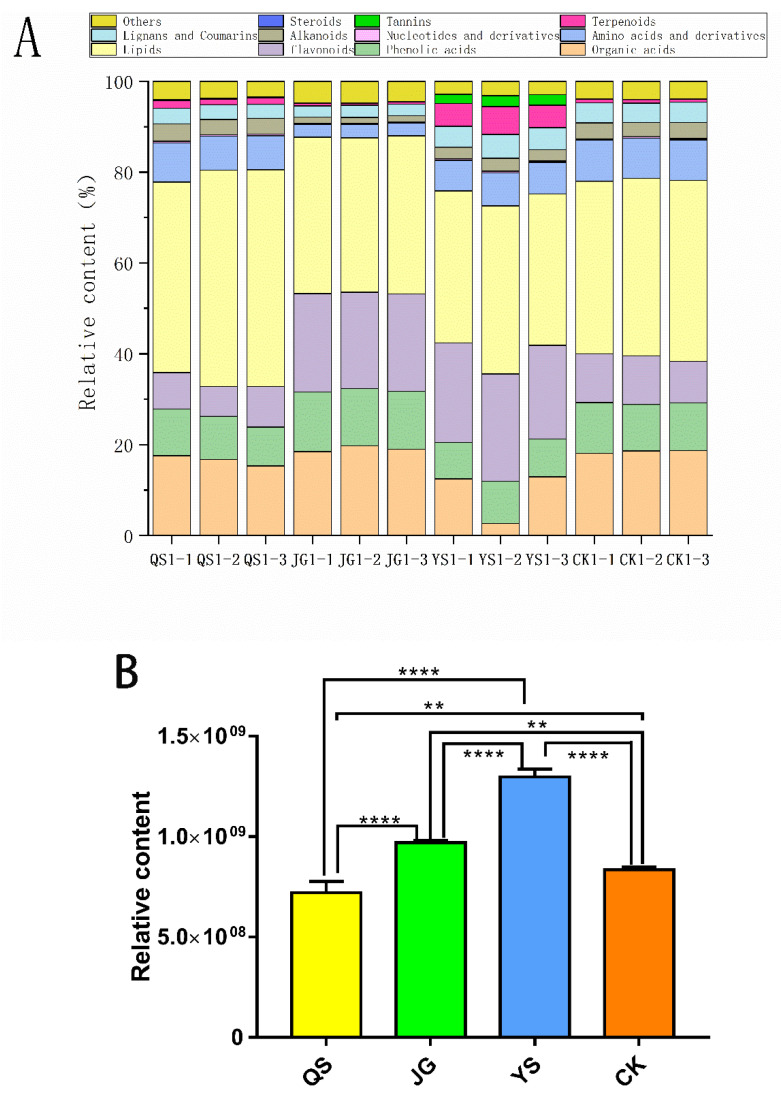
(**A**) Relative content stacked histogram of different metabolite categories. (**B**) One-way ANOVA analysis of total metabolite content of all samples in the experiment. The symbol with two five-pointed stars and the symbol with four five-pointed stars above the histogram indicate statistical significance at the level of 0.01 (*p* < 0.01) and 0.0001 (*p* < 0.0001), respectively.

**Figure 2 foods-12-00425-f002:**
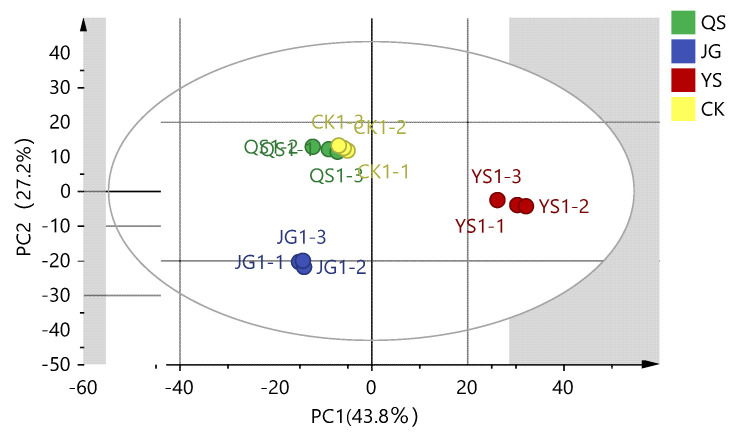
Score scatter plot of a principal component analysis composed of all metabolites (R^2^X[1] = 0.438, R^2^X[2] = 0.272).

**Figure 3 foods-12-00425-f003:**
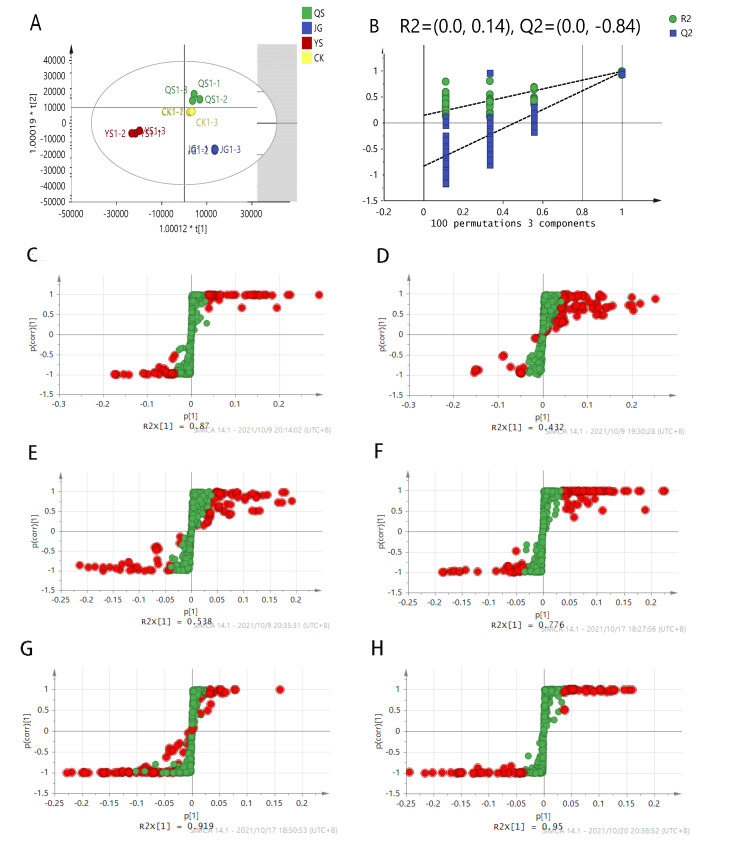
(**A**) Score scatter plot of the OPLS-DA model for all Laoxianghuang samples (R2X = 0.938, R2Y = 0.993, Q2 = 0.985). (**B**) Permutation test plot with 100 cycles. R^2^ represents the percentage of model matrix information; Q^2^ represents the predictive power of the original model. Loading S-plots (**C**–**H**) based on the OPLS-DA model for comparing samples in pairs, including QS–JG, QS–YS, JG–YS, QS–CK, YS–CK, and JG–CK.

**Figure 4 foods-12-00425-f004:**
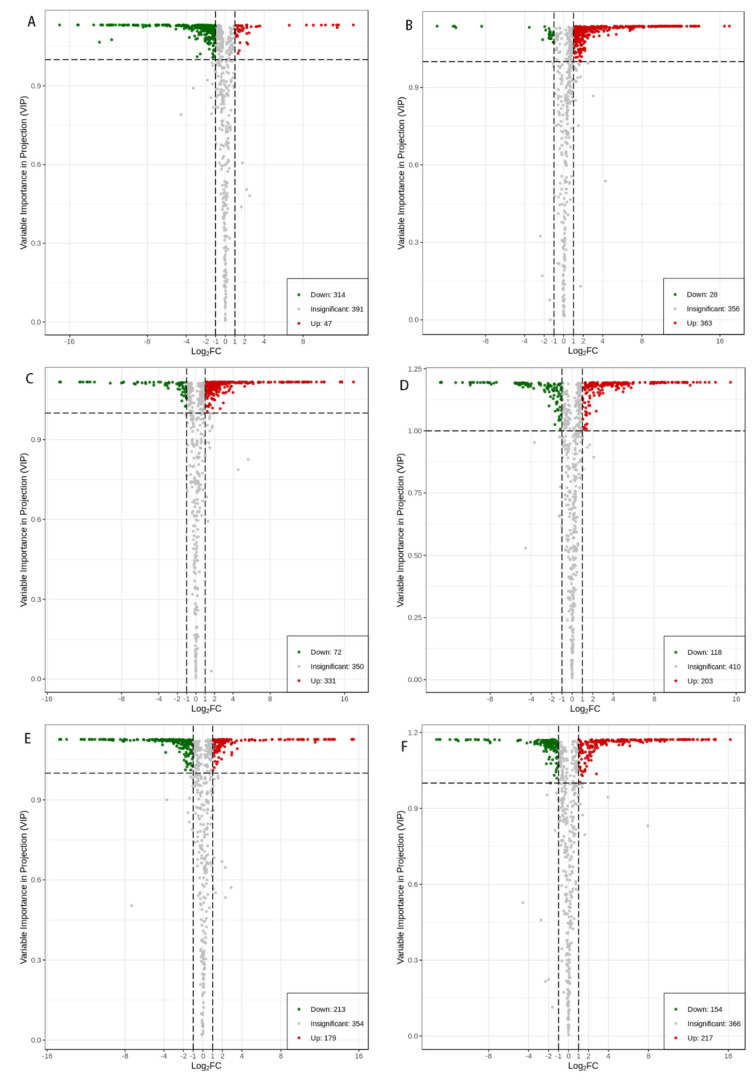
Volcano plots of differential metabolites presented in the two samples. (**A**) YS vs. CK; (**B**) YS vs. QS; (**C**) YS vs. JG; (**D**) CK vs. QS; (**E**) CK vs. JG; (**F**) JG vs. QS. The green points represent downregulated differentially expressed metabolites; the red points represent upregulated differentially expressed metabolites; and the black points represent metabolites that are detected but are not significantly different.

**Figure 5 foods-12-00425-f005:**
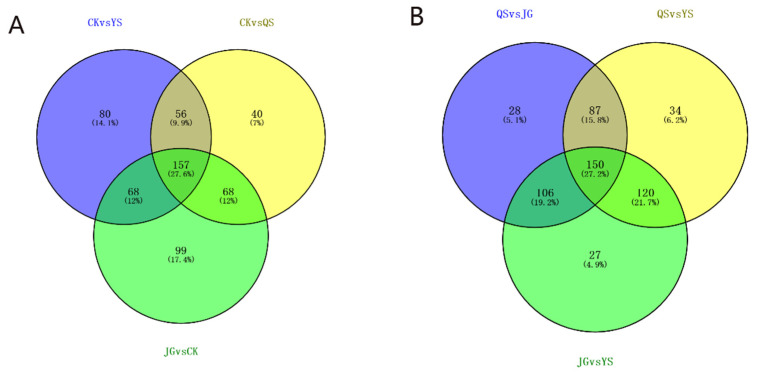
Venn diagrams of differential metabolites among different comparison groups. (**A**) CK vs. YS, CK vs. QS, and CK vs. JG; (**B**) QS vs. JG, QS vs. YS, and JG vs. YS.

**Figure 6 foods-12-00425-f006:**
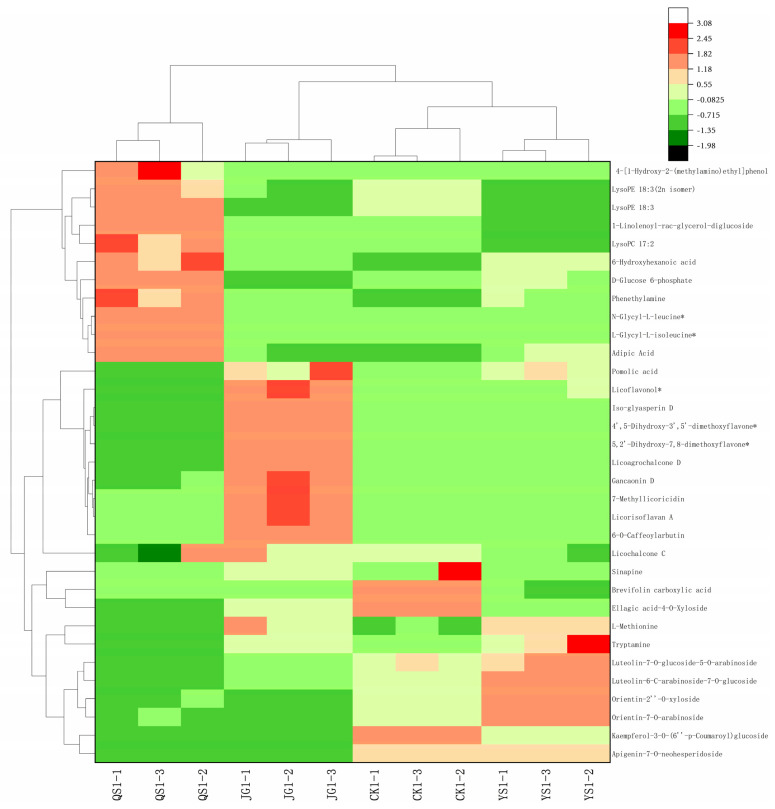
Heatmap of the hierarchical analysis of the 33 differential metabolites screened.

**Figure 7 foods-12-00425-f007:**
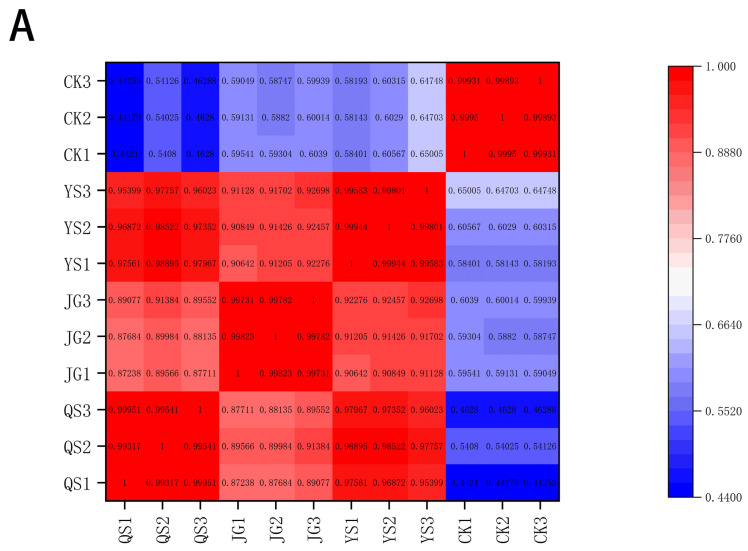
(**A**) Pearson correlation coefficient heatmap of the samples screened. (**B**) Pearson correlation analysis heatmap between the 33 differential metabolites screened.

**Figure 8 foods-12-00425-f008:**
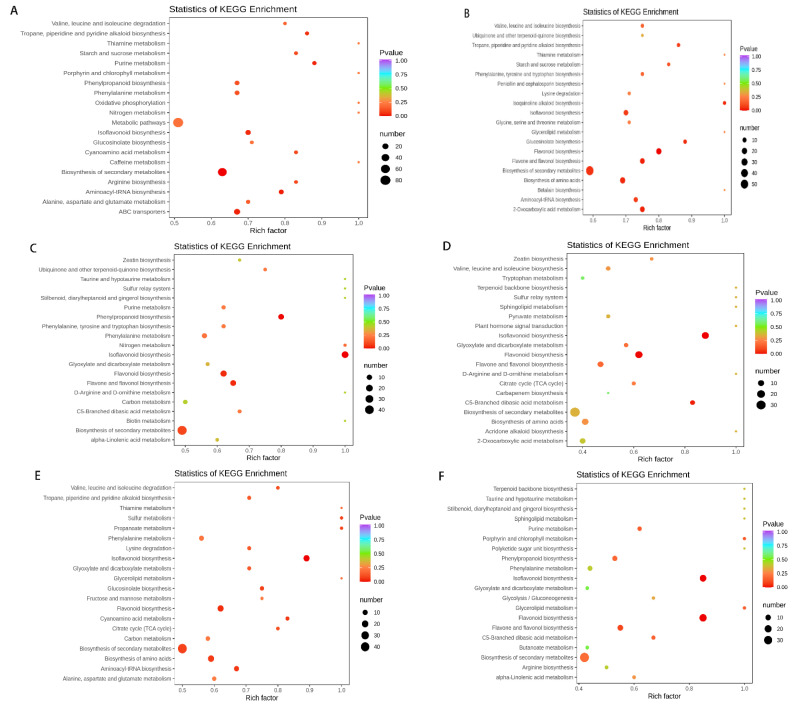
KEGG enrichment map of differential metabolites among the comparison groups (**A**–**F**), including JG vs. CK; JG vs. YS; QS vs. CK; QS vs. YS; JG vs. QS; and CK vs. YS. The abscissae represent the rich factor corresponding to each pathway. The higher the value, the higher the enrichment degree. The ordinates represent different metabolic pathways. The colour of the dot stands for the *p*-value; the redder the dot, the more significant the enrichment. The size of the dots represents the number of differential metabolites enriched in a given metabolic pathway.

**Table 1 foods-12-00425-t001:** Differential metabolites of the top 10 different multiples in different LXH samples as identified by VIP score and log2FC.

Group	Number	Compound	Substance Classification	Secondary Substance Classification	VIP Value	Fold Change	Log2(FC)	Type
	1	Apigenin-7-O-neohesperidoside	Flavonoids	Flavonoid	1.126417	44,938.5185	15.45566	Up
	2	Orientin-7-O-arabinoside	Flavonoids	Flavonoid carbonoside	1.126384	42,087.037	15.36109	Up
	3	Orientin-2″-O-xyloside	Flavonoids	Flavonoid	1.126273	39770	15.27939	Up
	4	Kaempferol-3-O-(6″-p-Coumaroyl)glucoside (Tiliroside)	Flavonoids	Flavonols	1.1264	11,743.6296	13.51959	Up
JG vs. YS	5	Luteolin-6-C-arabinoside-7-O-glucoside	Flavonoids	Flavonoid carbonoside	1.126356	10,588.5185	13.37021	Up
	6	Iso-glyasperin D	Flavonoids	Dihydroisoflavone	1.126365	0.00017205	−12.5049	Down
	7	Gancaonin D	Flavonoids	Isoflavones	1.126295	0.00006126	−13.99465	Down
	8	Licoflavonol *	Flavonoids	Flavonols	1.126353	0.00004100	−14.57389	Down
	9	7-Methyllicoricidin	Others	Others	1.126302	0.00003772	−14.69419	Down
	10	Licorisoflavan A	Others	Others	1.126226	0.00003746	−14.70413	Down
	1	Apigenin-7-O-neohesperidoside	Flavonoids	Flavonoid	1.137351391	127,896.2963	16.96461496	Up
	2	Licochalcone C	Flavonoids	Chalcones	1.137357845	90,041.85185	16.45830811	Up
	3	Luteolin-6-C-arabinoside-7-O-glucoside	Flavonoids	Flavonoid carbonoside	1.13735939	14,403.7037	13.81415221	Up
	4	Luteolin-7-O-glucoside-5-O-arabinoside	Flavonoids	Flavonoid	1.137319003	14,317.40741	13.80548265	Up
	5	Ellagic acid-4-O-Xyloside	Tannins	Tannin	1.137274445	11,843.33333	13.53178757	Up
QS vs. CK	6	L-Methionine	Amino acids and derivatives	Amino acids and derivatives	1.136742305	0.00296228	−8.399076127	Down
	7	Synephrine; 4-[1-Hydroxy-2-(methylamino)ethyl]phenol	Alkaloids	Amphetamine alkaloids	1.131501161	0.000475462	−11.03838303	Down
	8	6-Hydroxyhexanoic acid	Organic acids	Organic acids	1.136815382	0.000436773	−11.16082852	Down
	9	Phenethylamine	Alkaloids	Alkaloids	1.136818257	0.000391197	−11.31981845	Down
	10	Adipic Acid	Organic acids	Organic acids	1.137304026	0.000125694	−12.9578007	Down
	1	6-O-Caffeoylarbutin	Phenolic acids	Phenolic acids	1.171937206	75,198.14815	16.19840951	Up
	2	Licorisoflavan A	Others	Others	1.171857338	26,692.22222	14.7041318	Up
	3	7-Methyllicoricidin	Others	Others	1.171928468	26,508.88889	14.69418858	Up
	4	Licoagrochalcone D	Flavonoids	Chalcones	1.172028906	26,098.51852	14.67168029	Up
	5	Licoflavonol *	Flavonoids	Flavonols	1.171994179	24,388.14815	14.57389259	Up
QS vs. JG	6	Phenethylamine	Alkaloids	Alkaloids	1.171756278	0.000391197	−11.31981845	Down
	7	D-Glucose 6-phosphate	Others	Saccharides and alcohols	1.171821987	0.00026668	−11.87260363	Down
	8	L-Glycyl-L-isoleucine *	Amino acids and derivatives	Amino acids and derivatives	1.171979721	0.0001353	−12.85155386	Down
	9	Adipic Acid	Organic acids	Organic acids	1.172037319	0.000125694	−12.9578007	Down
	10	N-Glycyl-L-leucine *	Amino acids and derivatives	Amino acids and derivatives	1.172008186	0.000105116	−13.2157296	Down
	1	Adipic Acid	Organic acids	Organic acids	1.131895001	9257.296296	13.17637518	Up
	2	6-Hydroxyhexanoic acid	Organic acids	Organic acid	1.131963506	3209.592593	11.64817447	Up
	3	Tryptamine	Alkaloids	Plumerane	1.122093959	2937.077778	11.52016576	Up
	4	Phenethylamine	Alkaloids	Alkaloid	1.131758859	2803.37037	11.45294664	Up
CK vs. YS	5	Licoagrochalcone D	Flavonoids	Chalcone	1.131552846	2166.185185	11.08094087	Up
	6	Sinapine	Alkaloids	Alkaloid	1.066109475	0.00012731	−12.93936631	Down
	7	Ellagic acid-4-O-Xyloside	Tannins	Tannin	1.131894356	0.00008443	−13.53178757	Down
	8	LysoPE 18:3(2n isomer)	Lipids	LPE	1.131980975	0.00002816	−15.11581736	Down
	9	LysoPE 18:3	Lipids	LPE	1.131983277	0.00002707	−15.1724333	Down
	10	Brevifolin carboxylic acid	Phenolic acids	Phenolic acid	1.131925149	0.0000074284	−17.03851736	Down
	1	Apigenin-7-O-neohesperidoside	Flavonoids	Flavonoid	1.195389305	44,938.51852	15.45566494	Up
	2	Licochalcone C	Flavonoids	Chalcone	1.19539782	16,299.62963	13.99255156	Up
	3	Luteolin-6-C-arabinoside-7-O-glucoside	Flavonoids	Flavonoid carbonoside	1.195306886	10,588.51852	13.37021313	Up
QS vs. YS	4	Luteolin-7-O-glucoside-5-O-arabinoside	Flavonoids	Flavonoid	1.195286929	8024.925926	12.97027236	Up
	5	Pomolic acid	Terpenoids	Triterpene	1.195022849	5758.259259	12.49141703	Up
	6	LysoPC 17:2	Lipids	LPC	1.194661579	0.000600374	−10.70185192	Down
	7	Brevifolin carboxylic acid	Phenolic acids	Phenolic acids	1.194119719	0.000378969	−11.3656339	Down
	8	1-Linolenoyl-rac-glycerol-diglucoside	Lipids	Free fatty acids	1.195229758	0.000145036	−12.75130384	Down
	9	LysoPE 18:3	Lipids	LPE	1.194872468	0.000142903	−12.77268114	Down
	10	LysoPE 18:3(2n isomer)	Lipids	LPE	1.194628338	0.000133517	−12.8706858	Down
	1	Apigenin-7-O-neohesperidoside	Flavonoids	Flavonoid	1.116814	127,896.296	16.96461	Up
	2	Orientin-7-O-arabinoside	Flavonoids	Flavonoid carbonoside	1.116797	53,208.5185	15.69937	Up
	3	Kaempferol-3-O-(6″-p-Coumaroyl)glucoside (Tiliroside)	Flavonoids	Flavonol	1.11681	52,504.8148	15.68016	Up
	4	Orientin-2″-O-xyloside	Flavonoids	Flavonoid	1.116763	47,237.037	15.52763	Up
JG vs. CK	5	Luteolin-6-C-arabinoside-7-O-glucoside	Flavonoids	Flavonoid carbonoside	1.116777	14,403.7037	13.81415	Up
	6	5,2′-Dihydroxy-7,8- dimethoxyflavone *	Flavonoids	Flavonoid	1.116659	0.00021977	−12.15172	Down
	7	4′,5-Dihydroxy-3′,5′-dimethoxyflavone *	Flavonoids	Flavonoid	1.116652	0.00017971	−12.44203	Down
	8	Iso-glyasperin D	Flavonoids	Dihydroisoflavone	1.116734	0.00017205	−12.5049	Down
	9	Licoflavonol *	Flavonoids	Flavonols	1.116716	0.00004100	−14.57389	Down
	10	Licoagrochalcone D	Flavonoids	Chalcones	1.116758	0.00003831	−14.67168	Down

*: the substance has isomers, but the mass spectrometry database can’t tell which isomers it is.

## Data Availability

Data is contained within the article.

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
