# Peer review of "Identification and Analysis of Metabolites That Contribute to the Formation of Distinctive Flavour Components of Laoxianghuang"

_foods, 2023, doi:10.3390/foods12020425_

Round 1

Reviewer 1 Report

The manuscript entitled “Identification and analysis of metabolites that contribute to distinctive flavor components formation of the pickled bergamot (Citrus medica L. var. sarcodactylis Swingle Laoxianghuang“, conforms to the aims and the scope of the Journal. The paper is well written. The subject is interesting not only from scientific point of view, but also for its practical value. The manuscript aims to identify and screen differential metabolites from four Laoxianghuang samples collected from different regions in the Chaoshan area through untargeted metabolomics analysis based on UPLC‒MS/MS.

I suggest minor revision of manuscript.

Page 2 line 52

The uppercase of Rutaceae should be used.

Page 3 line 118

The examined samples are divided into 4 groups, not 3 as written in the text. Authors should indicate on what principle these 12 samples are divided into 4 groups.

Page 5 line 208

The correct Figure should mention in the text. ( Fig. 1 not Fig. 3c)

Page 6 line 239

The authors should write Fig. 2 caption.

Page 6 line 246

The correct Figure should mention in the text. ( Fig. 3A not Fig. 1 A)

Page 8 line 314 to 320

The text with Figures caption 4 and 5 should be deleted, because it is given in the correct places further down.

Page 16 line 445

The uppercase of Rutaceae should be used.

Author Response

Point 1 :Page 2 line 52, The uppercase of Rutaceae should be used.

Response 1: The original "rutaceae" is changed into "Rutaceae".

Point 2: Page 3 line 118 ,The examined samples are divided into 4 groups, not 3 as written in the text. Authors should indicate on what principle these 12 samples are divided into 4 groups.

Response 2 : "3" is changed into "4. I feel sorry for the little mistake.

Point 3: Page 5 line 208,The correct Figure should mention in the text. ( Fig. 1 not Fig. 3c)

Response 3 : "Fig. 3c" is changed into "Fig. 1".

Point 4: Page 6 line 239,The authors should write Fig. 2 caption.

Response 4: “(Fig. 2)” has been added after "PC2" in Page 6 line 241.

Point 5: Page 6 line 246,The correct Figure should mention in the text. ( Fig. 3A not Fig. 1 A)

Response 5: "Fig. 1A" is changed into "Fig. 3A.

Point 6 : Page 8 line 314 to 320,The text with Figures caption 4 and 5 should be deleted, because it is given in the correct places further down.

Response 6: The text with Figures caption 4 and 5 has been deleted.

Point 7: Page 16 line 445,The uppercase of Rutaceae should be used.

Response 7: “rutaceae” has been changed into “Rutaceae”.

Reviewer 2 Report

The manuscript titled "Identification and analysis of metabolites that contribute to distinctive flavor components formation of the pickled bergamot(Citrus medica L. var. sarcodactylis Swingle) Laoxianghuang" highlights some interesting metabolites quantified in the Laoxianghuang sample. Below are suggestions for authors to consider and a PDF has been attached for minor comments:

1) The title needs to be revised 

2) In the method section: Since the authors prepared their own sample, its should be clear how it was coded. They make mention of fresh sample in some parts of the manuscript and this again is not very clear as the samples were all stored for 1 year at room temperature. Again, there is no mention of how long the purchased samples were before being stored for an additional year, this should be included in the methods section. 

3) Results: The result section is very long and in some makes reference to published papers. Due to the lengthiness of the section, the readers might lose interesting results. There is also a need to include footnote information of the samples, particularly the one that was prepared by the authors since it acts as a control in this case. 

4) Some of the figures need to be saved in a higher resolution. 

5) Discussion was well prepared and straight to the point, might need to use the same writing strategy for the result section. 

Overall, the concept of the manuscript is good and will add value in the scientific community. 
